**Anoma Research Topics** | TECHNICAL REPORT

# Slow Games

**D Reusche[a], Christopher Goes[a], and Nicolas Della Penna[a]**

[a]Heliax AG

**\*** E-Mail: d@heliax.dev

## Abstract

Motivated by decentralized permissionless protocols that are ultimately backed by social consensus, which can only perceive and act much slower than the service provisioning, we study what we term a Slow Game; a type of principal-agent problem, in which the agent acts as operator of a service and the principal as a regulator, which sets and attempts to enforce policies on the service being provided. The regulator is slower acting and measuring than the operator, which introduces uncertainty depending on the difference in speed. In this publication we introduce a framework inspired by lossy compression problems to model this type of game, as well as present results from simulations of a minimal example.

**Keywords:** mechanism design ; distributed systems ; principal-agent problem ; lossy compression

## Contents

## 1. Introduction

**1.1. Conceptual framework.** An instance of the slow game consists of, at minimum:

1. A fast agent $f$ (which might be a coordinated group) taking actions. The identity of the fast agent, their action space, and costs/rewards to taking particular actions are specific to each instance.

2. A slow agent $s$ (which might be a coordinated group) taking measurements $m$. The identity of the slow agent, the measurements which can be taken, how frequently they can be taken, and how much they cost to take are specific to each instance.

3. A world model $w$ (which may or may not be fully known) which determines how the actions taken by the fast agent affect the measurements taken by the slow agent (often over time). The nature of the world model (and how much of it is known) is specific to each instance.

4. A regulatory mechanism $r$ through which the slow agent can reward or punish the fast agent, depending on the measurements which they take over time. The nature of the possible rewards and punishments is specific to each instance.

5. A target world profile $t$ chosen by the slow agent (often changing over time). This target profile may include actions taken by the fast agents, measurements taken by the slow agents, or in-between (inferable) variables of the world state. The type of the target world profile is specific to each instance, and the value is typically an input to the system over time.

The characteristic questions for a slow game instance are:

Given the action space and costs/rewards of the fast agent, the measurement space, frequencies, and costs of the slow agent, the (possibly uncertain) world model, and the available regulatory mechanism:

1. Can a policy $p$ be crafted which will achieve the target world profile in incentive-compatible equilibrium?

2. What is that policy?

3. What is the deviation between the reward profile of the actions which best maximize the target world profile, and the reward profile of the actions which best maximize the fast agent's returns? This could be called something like slack (or extractable value - this is a sort of generalized MEV).

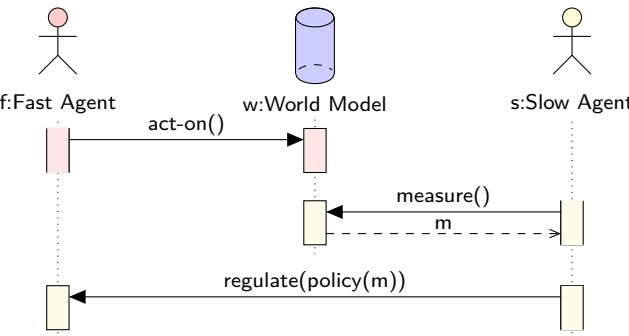

### 1.2. Examples.

**1.2.1. Controller selection in Anoma.** - Fast agent: controller in question, who can choose what fees to charge, and which transactions to possibly censor. The controller's reward is the fees paid, and possibly side rewards (bribes) for censorship. - Slow agent: users submitting transactions to the controller in question, who can measure the fees charged, and can measure over time whether particular transactions are being censored. - World model: fees are directly measurable; censorship is probabilistically measurable over time (since we also assume unreliable network conditions). - Regulatory mechanism: users can decide whether or not to pay fees, and they can switch controllers, which reduces future rewards for the controller to zero. - Target world profile: controller charges fees not more than a fixed margin above its operating costs and what would be needed to clear the market, and controller does not censor transactions.

### 1.3. Example: Solver selection in Anoma.
- Fast agent: Solver in question, who can choose to accept or not accept particular intents and to exploit slack (price differences between intents) or to return slack back to users. - Slow agent: Users submitting intents to the solver , who can measure (over time and by comparing with each other) whether the solver is censoring intents and how much slack is being returned to users. - World model: slack (MEV) return and censorship are probabilistically measurable over time (since we also assume unreliable network conditions). - Regulatory mechanism: users can decide whether or not to keep sending intents to this particular solver, which reduces future rewards for the solver to zero. - Target world profile: solver exploits slack not more than a fixed margin above its operating costs and does not censor intents.

**1.3.1. Delegated governance systems.** - Fast agent: governance delegates, who can make particular decisions more for their own benefit or more for the benefit of a public (slow agent). - Slow agent: voters, who can measure which decisions are made, or at least their impacts. - World model: decisions made impact the state of the world (very general). - Regulatory mechanism: varies, often voting out particular delegates on a periodic basis, sometimes also emergency referenda. - Target world profile: general happiness and stability.

## 2. Lossy compression model

**2.1. Model description.** We assume that the difference in speed between operator and regulator leads to only lossy observations of operator actions (or outcomes thereof) being possible on the regulator side: We call this difference in speed the **speed factor**, the loss induced by it **dropout**.

**Example:** If the operator acts 10 times within an interval, but the regulator can only measure 2 times, only 20% of the signal can be observed, the other 80% being dropout. The speed factor of the regulator in this case is 0.2.

Since we are interested in a quantitative analysis of how feasible it is for regulators to detect out-of-policy behaviors enacted by operators under uncertainty as described above, we take inspiration from lossy compression research, especially the concept of **perceptual quality**[1]:

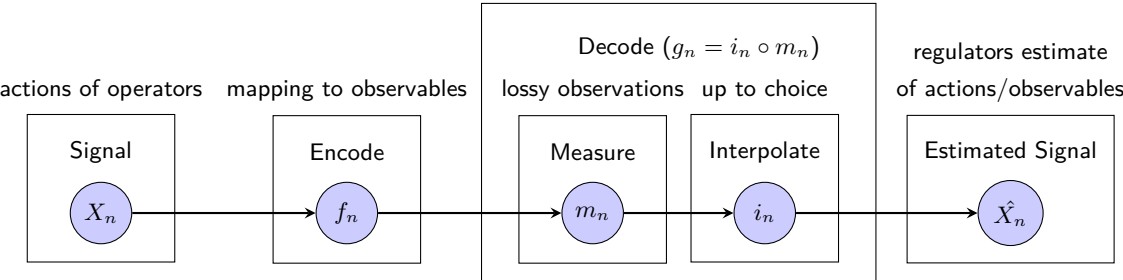

**Fig. 1:** Lossy compression model for operator ⇔ regulator interactions

For Fig. 1, $d(X_n, \hat{X}_n)$ describes the **perceptual divergence** of a signal and its estimate, with $d = 0$ meaning that signal and estimate are fully equal. Using the wasserstein metric[2] as a divergence measure, we can quantify the similarity between the distribution pairs of a signal and its estimate.

The **perceptual quality** of an estimate quantifies how likely it seems to correspond to a valid signal, in our case a set of in-policy actions, independent of what the signal actually was.

### 2.1.1. Game formulation and knowledge requirements.
To build the bridge from perceptual quality to the slow game problem, we can rephrase the last statement as follows: **How much could an operator cheat, while still producing a signal of which the estimate convinces the regulator of in-policy behavior, given the regulators lossy observations?**

To answer this question, we need to know the following:

**1) The speed factor at which observations happen.** See 2.1.4.

**2) A baseline distribution implementing in-policy behavior.** This is used as a signal to compute estimates with simulated, slowness induced, loss and in turn the slowness induced measurement error (**pure error**). Requirements for this are knowledge of the policy, as well as distributions the actions are drawn from (e.g. preferences or constraints of participants). Given these, we can e.g. derive closed form models, or produce empirical distributions via simulation.

**3) Optionally, information about the cheating mechanism of the operator.** More specifically, information about the conflation of pure error and cheating mechanism for out-of-policy behaviors at known speed factors.

**Note:** In many cases, the operator will have access to the regulators knowledge, but more rarely the reverse be the case: A service provider can be their own user easily, but a user not always the provider of a service they are using.

We now introduce notation for these relevant types of signals[3]: Let $G_n$ be a baseline of "good" in-policy behavior, $B_n$ an example of "bad" out-of-policy behavior, $S_n$ some observed sample behavior, with $\mathbb{S} = \{s \in \mathbb{Q} \mid 0 \leq s \leq 1\}$ being a family of speed factors $s$ and $d_s(\cdot, \cdot)$ the divergence measure of signal to estimate at a given speed factor $s$.

For a given speed factor $s$, we define the following metrics:

- $e_s(G_n) = d_s(G_n, \hat{G}_n)$ the **pure (slowness induced) error** of observations by the regulator. It tells us how close the estimate of a known good signal is to the signal itself.

- $c_s(B_n) = d_s(B_n, \hat{B}_n)$ the **cheating prior**. It tells us how close the estimate of a *specific* bad signal is to the signal itself, *including* interactions of the cheating mechanism with the pure error (the above mentioned conflation).

- $o_s(G_n, \hat{S}_n) = d_s(G_n, \hat{S}_n)$ the **observed divergence (from baseline)**. It gives a distance between the estimate of some observation from the estimate of a known good signal.

- $x_s(G_n, \hat{S}_n) = o_s(G_n, \hat{S}_n) - e_s(G_n)$ the **excess divergence**. It tells us how much of the observed divergence is not explained by pure error.

**Note:** For ease of exposition, we look at observables $O_n = f(X_n)$, instead of the actions/signal $X_n$. In general actions might not be observable at all, i.e. there is never access to signal samples. Thus, policies should be defined over observables $O_n = f(X_n)$, and estimates $\widehat{O_n}$ should be computed accordingly, unless the mapping between an observable and the latent variable modelling the signal is clear and policing the actions directly is desirable.

---

[1] Blau and Michaeli (2019)
[2] We use the wasserstein metric with square euclidean distance, instead of e.g. KL-Divergence, because it is a proper metric, i.e. gives us interpretable values everywhere.
[3] We assume that an estimate can be computed for any signal that is available, but not the other way around.

**2.1.2. Crafting incentive structures.** Since we want to incentivize in-policy behavior, we need to define a reward/punishment mechanism to achieve that.

For example, assuming some base reward $R_b$ and operating cost $C_o$, we could try compute a weighting factor $w$, which depends on how far we deem the operators behavior to be away from in-policy behavior, while taking the uncertainty of our measurements at a given speed factor $s$ into account.

To do that, we can use the measures for pure error, cheating prior and excess divergence from above to define **payoff weights**. In general, we subtract the excess divergence from the respective prior (**Note:** The pure error can be seen as a prior with no information about conflation with cheating distributions):

- In case we only know a baseline $G_n$:

$$w_s^G(G_n, \hat{S}_n) = e_s(G_n) - x_s(G_n, \hat{S}_n) \tag{1}$$

- In case we also know the cheating prior $B_n$:

$$w_s^B(G_n, B_n, \hat{S}_n) = c_s(B_n) - x_s(G_n, \hat{S}_n) \tag{2}$$

This gives us reward weights $w_s$, which we can use directly in our payoff function. Then **payoff** for $S_n$ at speed factor $s$, derived from a good baseline is: $p_s^G(\hat{S}_n) = (R_b - C_o) \cdot w_s^G(G_n, \hat{S}_n)$. If cheating priors are available: $p_s^B(\hat{S}_n) = (R_b - C_o) \cdot w_s^B(G_n, B_n, \hat{S}_n)$. When not explicitly denoted, $p_s$ can be either $p_s^G$ or $p_s^B$.

**2.1.3. Regret formulation.** To check how well we incentivize in-policy behavior with the payoff function from above, we calculate **regret** for all parameter sets of the cheating mechanism which are simulated per speed factor. E.g. if the cheating mechanism samples from a binomial distribution $B(10, p_i)$ with $p_i \in \{0.1, 0.2, ..., 1\}$, we receive for each $p_i$ a different corresponding $\hat{S}_{n_i}$. For now we assume that lower values for $p$ mean less cheating, with $p = 0$ being no cheating at all. Regret then is the payoff for some $p_i$ subtracted from the best payoff over all $p_i$:

$$r_s(p_i) = \left( \max_{\forall p_i} p_s(\hat{S}_{n_i}) \right) - p_s(\hat{S}_{n_i}) \tag{3}$$

So if we want the dominant strategy to be in-policy behavior, regret should be minimized at $r_s(0)$ for any given $s$.

**Note:** When using weights $w_s$, we can observe empirically in the experiment explained below, that in-policy behavior minimizes regret for the operator, with reward being positive. Further work will need to show if this generalizes.

**2.1.4. Speed Games.** To determine the speed factor between regulator and operator, another game can be played, which we sketch here: Since a lower ratio of regulator:operator speed leads to more leniency of the regulator in our setting, the incentive of the operator is to convince the regulator of as high a speed as possible.

Assuming the regulator incurs some cost $c(f)$ for measuring at frequency $f$, the operator could offer (a part of) this cost to compensate the regulator for the process of proving their capability to operate at $f$.

Actual operation after the proof could take place at a lower frequency, but depending on measurement protocols the regulator might detect that and adjust the speed factor in its models, plus some additional punishment, e.g. in case some operating speed is agreed on.

To access a wider range of trade-offs between measurement cost and strength of deterrence, the regulator can e.g. randomize the measurement frequency.

**2.1.5. Interpolation and heavy tails.** Since dropout leaves us with incomplete data, we have the choice of interpolation scheme, e.g. replacing missing values with the mean of the interval, or using linear, polynomial or spline interpolation.

This has implications for which types of policies are feasible to (approximately) enforce: If payoff for defection is distributed in subgaussian fashion, i.e. "small" amounts of value can be extracted in a lot of events, interpolation will introduce tolerable error. If defection payoff is distributed in very heavy tailed ways, i.e. a lot of value can be extracted in very rare events, interpolation error is potentially very large.

Because of this, setups with subgaussian defection payoff are preferable. E.g. is certain choices of constraints for the system can be chosen that smooth out the distributions, that is preferable.

## 3. Example: Two player thermostat

Let us now work out (and implement simulations for[4]) a minimal example of a slow game using the above approach. For that, we pick a two player thermostat:

---

[4]The implementation can be found here: https://github.com/anoma/slow-game-research

**3.1. Game Model.** We have the following roles and objects:

- **Outside**, which has fluctuating temperature (drawn once per timestep from a discrete uniform distribution $\mathcal{U}(10, 32)$) and influences the temperature of a room.

- A **room**, which is supposed to be kept within in a certain range of temperature.

- An **operator** which

    - heats and cools the room to control its temperature.

    - tries to maximize its reward for the service provided (i.e. is a profit maximizing actor), using a stochastic cheating mechanism to cool or heat slightly less than necessary (drawing from binomial distribution $D_c = B(n, p)$).

- A **regulator**, which

    - sets the policy for the temperature bounds of the room. Here, the range is [18, 25].

    - tries to verify policy adherence of the operator.

    - rewards or punishes operator depending on the degree of adherence to policy.

The reward is computed by setting a heating/cooling budget $R_b$ for a period with $TS$ timesteps, and giving all unspent budget to the operator as a base reward. Heating or cooling by one degree costs one unit of the budget.

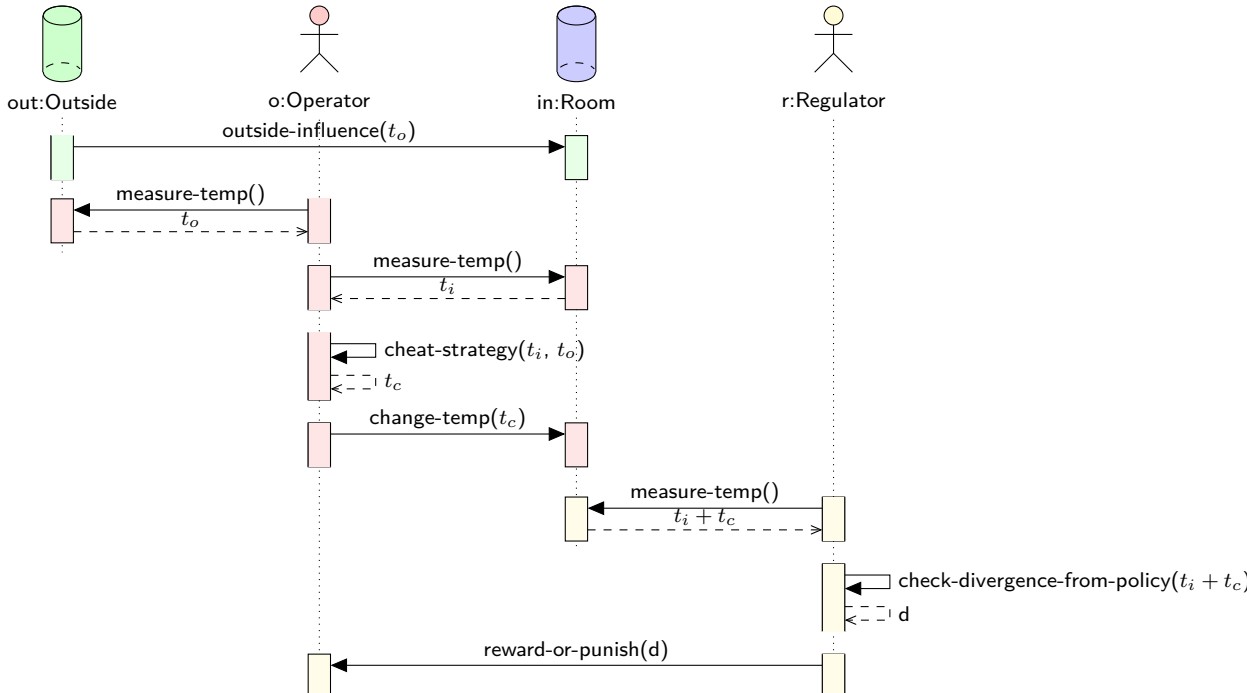

Here, the operator (red) and regulator (yellow) actions, happen at different frequencies $f_o$ and $f_r$ with $f_r < f_o$ and $f_o =$ once per time step. Outside influence can be continuous in time, but no change faster than maximal operator measurement frequency is relevant for our model.

**Note:** Assuming instantaneous temperature exchange between the outside and the room (e.g. the room has no insulation), the operator can omit either one of the temperature measurements.

Given the above model, with $T_c = \sum t_c$ over all timesteps, the payoff functions for the operator are:

$$p_s^G(\hat{S}) = (40000 - T_c) - w_s^G(G_n, \hat{S}) \qquad [4]$$

$$p_s^B(\hat{S}) = (40000 - T_c) - w_s^B(G_n, B_n, \hat{S}) \qquad [5]$$

For plots of $w_s^G$ and $w_s^B$, see subfigures 2.2 and 2.3 below.

**3.2. Empirical Analysis.** To get an intuition for how our example game plays out, given the above model and basic incentives, we simulate experiments and perform empirical analysis on it. We run experiments with $D_c = B(10, p_i), p_i \in \{0, 0.1, 0.2, ..., 0.9, 1\}$, with $S = 10000$, $R_b = 40000$, and interpolation replacing missing values with the mean of available data.

The subfigures below show the following:

**1.2** shows the pure error. (Error for $p = 0$ repeated along $p_i$.)

**1.3** shows the cheating prior. Since we know the full signal, we can compute $d_s$ for every signal/estimate pair.

**2.1** shows excess divergence. We don't assume to know any signal apart from the baseline, only observed estimates.

**2.2** payoff weights $w_s^G$ derived from pure error.

**2.3** payoff weights $w_s^B$ derived from cheating priors.

**3.1** base reward for operator w/o payoff weights.

**3.2** reward weighted entrywise by 2.2. according to Eq. 1.

**3.3** reward weighted entrywise by 2.3. according to Eq. 2.

**4.1** regret w/o reward weighting.

**4.2** regret corresponding to 3.2.: in-policy behavior is dominant strategy up to roughly 0.65 speed factor.

**4.3** regret corresponding to 3.3.: in-policy behavior is dominant strategy in all speed regimes.

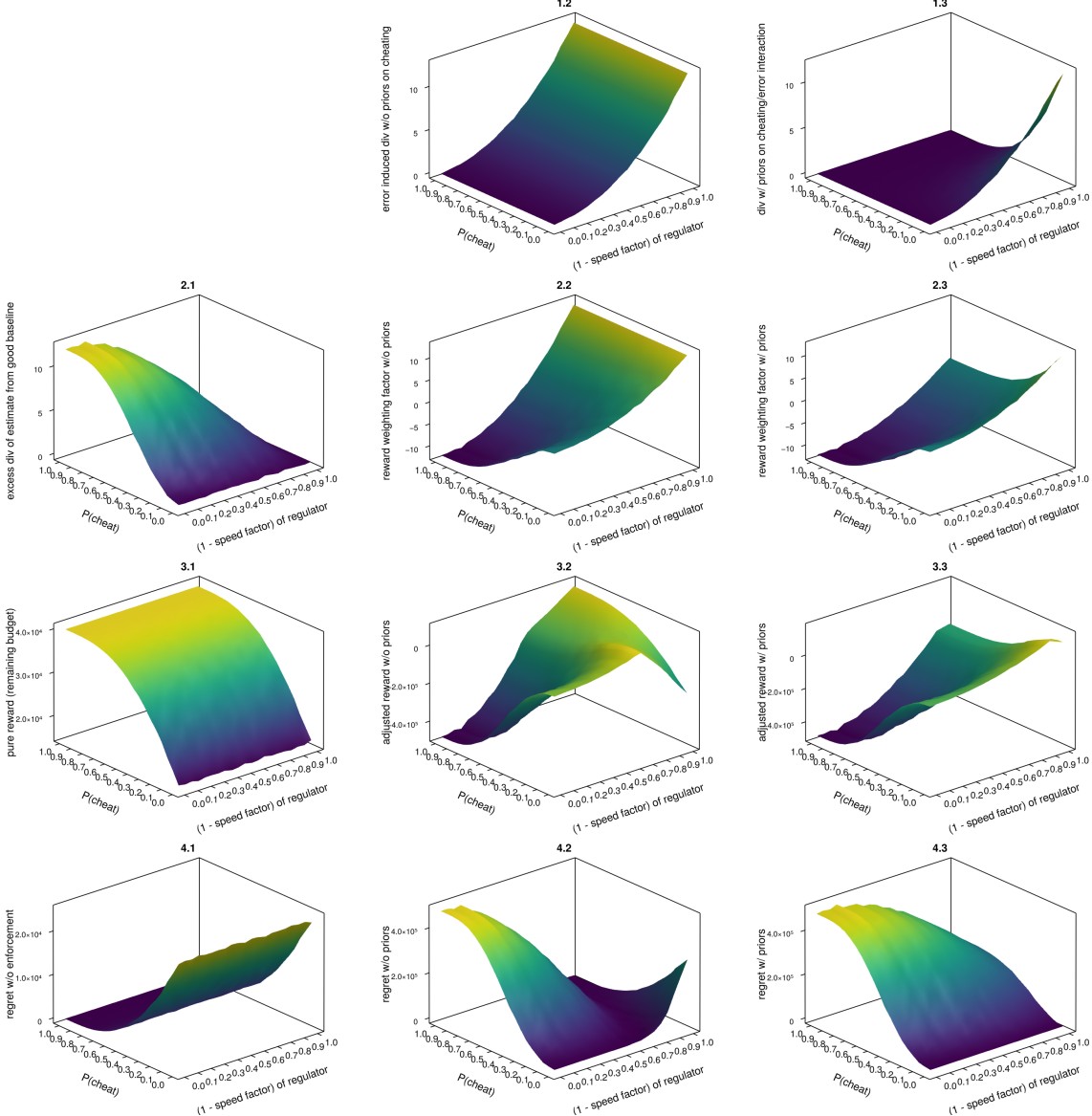

**Fig. 2:** Experiments with $D_c = B(10, p_i), p_i \in \{0, 0.1, 0.2, ..., 0.9, 1\}$

### 3.2.1. Discussion.
Looking at 1.2 and 1.3 we can see that knowing how the conflation of the cheating strategy with the pure error gives us tighter divergence information compared to only the pure error, especially in the regimes with both high speed difference and high cheating probability/

Subtracting excess divergence (2.1) from either of the above gives us different weighting surfaces for the reward, the result of which is shown in 3.2 and 3.3 respectively, with 4.2 and 4.3 being the corresponding regret formulations.

We can see negative reward payments, (i.e. punishment) in the low speed difference and high cheat regimes in both cases, causing high regret to the operator. The weights derived from the pure error result in rewards for cheating in the high speed difference regime though, i.e. the policy is too lenient.

The policy derived from the conflation (1.3) is tight enough in all speed regimes to incentivize in-policy behavior, as regret reliably increases together with cheating everywhere.

## A. Variation of cheating prior

We show another experiment, with different parameters for the cheating distribution: $D_c = B(3, p_i), p_i$ as above.

**1.2** shows how the pure error stays the same.

**1.3** shows how the conflation of pure error and cheating prior having a different shape.

**2.1** since in the excess divergence measurements, we only observe the estimates, this shape also changes.

**2.2-4.3** are analogous to Fig. 2, but derived from 1.3 and 2.1.

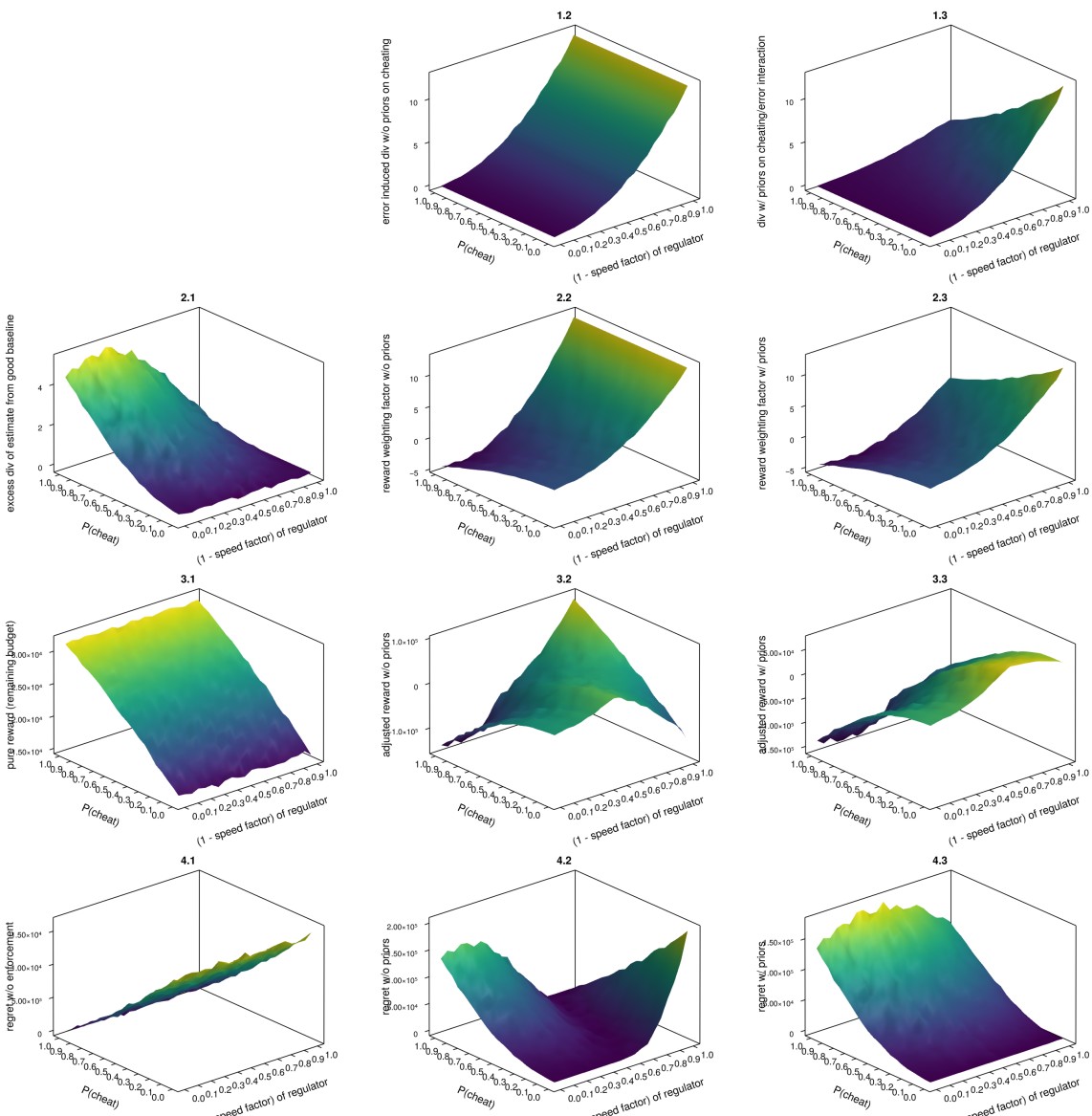

**Fig. 3:** Experiments with $D_c = B(3, p_i), p_i$ as above, to show cheating prior where pure error is conflated with a different distribution

## B. Acknowledgements

We thank the reviewers of the Agentic Markets Workshop at ICML 2024[5] for their helpful feedback on improving the exposition and refining this work.

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
