# OpenReview forum: "Slow Games"
_ICML.cc/2024/Workshop/Agentic_Markets — Agentic Markets @ ICML'24 Poster_

### Official Review · Reviewer_u93y · 2024-06-12
**Principal Agent Problem with slow Principal and Blockchain d**

**Rating:** 4
**Confidence:** 4

**Review:**

The paper presents a type of principal-agent framework, termed Slow Game,  where the agent is an operator of a service and the principal is the regulator that oversees the quality of the service provided by the operator by imposing rewards/punishment. It is described as repeated game.
The main coneptual contribution is that the principal takes action with smaller frequency compared to the agent, and this introduce uncertainty depending on the difference in speed.

**Comments - Questions**
- The paper is not anonymous
- There is no literature review. There is a very relevant paper and worth reading. "An Analysis of Intent-Based Markets" by Chitra et. al.
- Even though the report is inspired by decentralized systems, there is just a minimal example regarding intent markets that relates to the model.
- Could we cosider that
- In the example of the Two player thermostat where I was expecting the model to make more sense, neither the payoff function (i.e. world model) nor the reward/punishment function are explicitly defined. Moreover, in this example, I cannot see how the frequencies $f_r , f_o$ influence the payoff functions.
- There is not any explanation on the provided experimental results.

---

### Official Review · Reviewer_5TQZ · 2024-06-14
**The paper aligns with the workshop themes by addressing principal-agent problems in decentralized systems through a novel application of lossy compression. While the framework is innovative and insightful, improvements are needed in providing a holistic introduction, detailed simulation analysis, and discussing broader real-world applications.**

**Rating:** 7
**Confidence:** 3

**Review:**

The paper aligns well with the workshop themes, focusing on multi-agent systems, cooperative AI, and market design. It addresses principal-agent problems in decentralized systems, highlighting interactions between fast and slow agents.

**Strengths**
1. Agent Interaction and Security: Addresses coalition, cooperation, and commitments, relevant to agent interaction and security.
2. Novel Application: Introduces a novel application of lossy compression to economic interactions and agent dynamics.
3. Market Design: Provides insights into market dynamics and mechanism design.
4. Complexity of Autonomous AI Agents: Discusses agent interactions and complexity in financial markets, aligning with workshop themes.

**Suggested Improvements**
1. Introduction: The introduction dives directly into the details of the framework without providing a holistic overview of the problem. Start with a broader context to help readers understand the significance and scope of the research.
2. Conceptual Framework: In section 1.1 a diagram for the conceptual framework could enhance its impact by visually summarizing the key components and interactions.
3. Results Interpretation: In section 3.2 provide a more comprehensive analysis and interpretation of the simulation outcomes. Discuss any observed patterns and their implications for the framework.
4. Broader Real-World Applications:  The paper focuses on the theoretical framework but lacks discussion on potential real-world applications. Add a section discussing how "Slow Games" can be applied to various agentic markets, such as financial trading systems or supply chain management. Include more examples beyond the two-player thermostat game to demonstrate the framework's versatility. Examples from different domains, like financial markets or logistics, could illustrate the model's wide applicability.